# Japanese Encephalitis Virus Interaction with Mosquitoes: A Review of Vector Competence, Vector Capacity and Mosquito Immunity

**DOI:** 10.3390/pathogens11030317

**Published:** 2022-03-03

**Authors:** Claudia Van den Eynde, Charlotte Sohier, Severine Matthijs, Nick De Regge

**Affiliations:** 1Exotic Viruses and Particular Diseases, Sciensano, Groeselenberg 99, 1180 Brussels, Belgium; charlotte.sohier@sciensano.be (C.S.); nick.deregge@sciensano.be (N.D.R.); 2Enzootic, Vector-Borne and Bee Diseases, Sciensano, Groeselenberg 99, 1180 Brussels, Belgium; severine.matthijs@sciensano.be

**Keywords:** Japanese encephalitis virus, vector competence, vector capacity, vector immunity, virus–vector interactions, arboviruses

## Abstract

Japanese encephalitis virus (JEV) is a mosquito-borne zoonotic flavivirus and a major cause of human viral encephalitis in Asia. We provide an overview of the knowledge on vector competence, vector capacity, and immunity of mosquitoes in relation to JEV. JEV has so far been detected in more than 30 mosquito species. This does not necessarily mean that these species contribute to JEV transmission under field conditions. Therefore, vector capacity, which considers vector competence, as well as environmental, behavioral, cellular, and biochemical variables, needs to be taken into account. Currently, 17 species can be considered as confirmed vectors for JEV and 10 other species as potential vectors. *Culex tritaeniorhynchus* and *Culex annulirostris* are considered primary JEV vectors in endemic regions. *Culex pipiens* and *Aedes japonicus* could be considered as potentially important vectors in the case of JEV introduction in new regions. Vector competence is determined by various factors, including vector immunity. The available knowledge on physical and physiological barriers, molecular pathways, antimicrobial peptides, and microbiome is discussed in detail. This review highlights that much remains to be studied about vector immunity against JEV in order to identify novel strategies to reduce JEV transmission by mosquitoes.

## 1. Introduction

Japanese encephalitis virus (JEV) is a zoonotic mosquito-borne flavivirus (family Flaviviridae) that is maintained in a transmission cycle between the mosquito vectors and vertebrate hosts, mainly Ardeid birds (herons and egrets are considered natural reservoirs), pigs (amplifying hosts), and possibly bats. These vertebrate hosts produce high viremias [1], allowing mosquitoes to become infected when taking a blood meal (Figure 1). While JEV is generally considered to be a mosquito-borne disease, Ricklin et al. [2] recently demonstrated that direct virus transmission can also occur between pigs, via oronasal secretions. The epidemiological relevance of this finding is, however, unclear.

Birds of the family Ardeidae do not demonstrate clinical disease [3]. This is in contrast to pigs, an amplifying host, in which JEV can cause abortion or lead to mummified, weak, or stillborn piglets after infection of pregnant sows. Infected boars can become infertile upon infection. Humans, cattle, and horses are considered to be dead-end hosts, since JEV infection results in an insufficient viremia to infect naïve mosquitoes when taking a blood meal. Nevertheless, infection of these hosts can result in encephalitis, in combination with fever, tremors, convulsions, coma, and death [4]. In humans, and mostly in children [5], 1% of infected individuals will develop encephalitis, with a mortality rate in this group with disease symptoms of 20 to 30% [6]. JEV was first isolated in 1935 [7] and is a leading cause of viral encephalitis in Asia, with 30,000–50,000 human cases reported annually [8].

JEV is currently endemic in Australia (Torres Strait islands) and southeast and east Asia, including the temperate zone of northeastern China, Japan, and Korea [6] (Figure 2) and exists in five different genotypes. Genotypes one (G-I), two (G-II), and three (G-III) are found throughout Asia, genotype four (G-IV) in Indonesia, and genotype five (G-V) in Malaysia, China, and Korea [9]. G-III was the predominant genotype in Japan and Korea up to the 1990s [10]. A shift towards the dominance of G-I strains has, however, been recorded since 1995 [11]. G-III strains have also been detected outside of their endemic areas, e.g., in Italy and Angola [12].

A study by Oliveira et al. [13] identified a number of potential entry routes for JEV in the US, e.g., (1) entry through infected vectors by means of aircraft, ships, wind, or on imported tires; (2) importation of viraemic animals, e.g., pigs; (3) entry of viraemic migratory birds; (4) importation of infected biological materials; (5) importation of infected animal products; (6) entry of infected humans by globalization; and (7) importation/production of contaminated biological material, e.g., vaccines. However, since humans are considered to be dead-end hosts for JEV (exhibit only low levels of viremia), it is unlikely that infected humans would contribute to the spread of JEV. According to Oliveira et al., the most probable method of introduction is through the entry of infected adult mosquitoes via aircraft and ships/containers.

Upon introduction into non-endemic areas, JEV could then continue to be transmitted and possibly become established if competent vectors and suitable hosts are present. Competent vectors are mosquito species that have been shown to transmit JEV [14]. Competent vectors may be exotic or endemic mosquitoes. Invasive mosquitoes, e.g., *Aedes albopictus* and *Aedes japonicus*, are becoming more common and able to form permanent colonies in Europe, due to the current climate changes (warmer summers). On the other hand, indigenous mosquitoes may also be or may become (more) competent as a result of changing climatic factors, given that higher temperatures are known to increase the competence for flaviviruses [15] and shorten the extrinsic incubation periods (EIPs) [16]. Introduced infected mosquitoes could lead to infection of susceptible animals in these areas. Alternatively, infected viraemic animals could be imported. Subsequently, indigenous mosquitoes can become infected by taking a blood meal from these infected animals and transmit JEV if these species are competent. Vector competence studies should, thus, be carried out for mosquito species that are present in areas where JEV is not yet endemic, in order to evaluate which species could potentially transmit JEV in the event of an introduction.

Therefore, we reviewed the current knowledge on vector competence of mosquitoes for JEV and JEV detection in field-caught mosquitoes to get an idea of which species could have the highest vectorial capacity. Next, we also reviewed the available information on mosquito immunity against JEV in order to summarize the currently known underlying factors that influence the vector competence for this virus. Important factors of vector immunity are physical and physiological barriers, molecular pathways, antimicrobial peptides, and the vector microbiome.

## 2. Results

### 2.1. Mosquito Vectors of JEV: Vector Competence and Capacity

#### 2.1.1. JEV Detection in Field-Collected Mosquitoes

An initial systematic review of the literature has revealed that JEV has so far been detected in more than 30 mosquito species, belonging to the genera *Aedes*, *Anopheles*, *Armigeres*, *Coquillettidia*, *Culex,* and *Mansonia* (Table 1). Detection studies are often conducted on a large scale, where pools of field-collected mosquitoes are tested per species. Once the mosquito pools have been tested, information about the number of mosquitoes collected, the number of pools that tested JEV-positive, and the number of mosquitoes in each individual pool are used to calculate the estimated infection rate. There are a variety of methods to estimate infection rate. The most reported is the minimum infection ratio (MIR), which is the ratio of the number of positive pools to the total number of mosquitoes in the sample [17]. The MIR is often an underestimation, as it assumes that only one individual of the pool is positive, whereas multiple individuals of the pool could be positive [18]. Therefore, small-sized pools are preferred in order to obtain a more accurate estimate of the MIR. Besides the pool size, also the number of mosquitoes collected, and the virus detection method may influence the MIR. Six methods have been used for virus detection (see Table 1), e.g., plaque or hemagglutination inhibition (HI) and complement fixation (CF) assays, reverse transcription polymerase chain reaction (RT-PCR), intracerebral inoculation of mice, virus isolation on continuous cell lines, ELISA, and inoculation of *Toxorhynchites splendens* mosquito larvae (*Toxo*-IFA). While RT-PCR is the most sensitive and specific, only intracerebral inoculation of mice, virus isolation, and *Toxo*-IFA can differentiate between the infectious and the non-infectious virus, although with lower sensitivity. Consequently, these different methods make it difficult to compare across studies.

Using the data from 61 publications on the detection of JEV in field-collected mosquitoes, the MIR was calculated for 35 species. Differences in the total number of mosquitoes tested among studies ranged from 18 to 290,126. This partly explains the large differences in the MIR for JEV between the different species (from 0.0009 to 5.56%). If a comparison is made between those where larger numbers have been tested, it can be concluded that, for example, *Culex pipiens* (with MIR values from 0.01 to 0.54%) and *Culex tritaeniorhynchus* (MIR from 0.0009 to 1.01%) are often detected as JEV-infected in the field. Most studies do not differentiate between *Culex pipiens pipiens* and *Culex pipiens molestus*, therefore, in this review *Culex pipiens* refers to both, while *Culex pipiens pallens* is considered separately.

*Culex quinquefasciatus* was repeatedly found to be positive in Vietnam, although no MIR could be calculated for this study since the total number of tested specimens was not reported [70]. The detection of JEV in a specific field-collected mosquito species does not necessarily mean that this species is competent to transmit the virus to another host [81]. For a species to be considered competent, JEV needs to be able to disseminate in the vector after the blood meal and reach the saliva in order to be transmitted to other hosts. Table 2 gives an overview of field-collected mosquito species in which JEV has not been detected, despite screening efforts. The absence of field detection, however, cannot lead to the conclusion that these species are not JEV vectors. That would require additional studies, including vector competence studies, as described below. In several of the studies a very small number of mosquitoes was tested, e.g., three individuals for *Aedes aegypti* and one individual for *Aedes lineatopennis* [43], *Anopheles ludlowae,* or *Culex brevipalpis* [19], which precludes final conclusions.

#### 2.1.2. JEV Vector Competence Studies

Vector competence is defined as the intrinsic ability of a mosquito to acquire the pathogen, and subsequently transmit the pathogen to a new host [82]. This parameter can be determined based on laboratory experiments that determine the infection, dissemination, and transmission rates. These describe, respectively, the presence of the virus in the whole body of the mosquito (detection in the legs, wings, and/or mosquito heads) and the number of mosquitoes with viral particles in their saliva after infection [83]. Only those mosquitoes in which the virus reaches the saliva are considered to be competent mosquitoes. Where most studies determine the presence of the virus in the saliva by qPCR or virus isolation, actual transmission competence can be verified by allowing infected mosquitoes to feed on naïve animals and check for viremia and seroconversion in the host. A detailed overview of vector competence studies for JEV can be found in Table 3.

There are many variations in methodology between studies and differences in mosquito populations, which can influence the outcome of vector competence studies. From Table 3, it can be noted that differences in vector competence are reported between studies for the same mosquito species, e.g., the transmission ratio of 0% (New Zealand [84]) compared to 70% (UK [85]) for *Culex quinquefasciatus*. Populations differ genetically, depending on where they have been collected and how long the colony has been maintained in the laboratory [86]. Another influencing factor might be the viral strains used. For example, *Culex tritaeniorhynchus* showed higher viral titers in their saliva for G-III strains than for G-I and G-V [12]. However, in this study, no significant differences were recorded in transmission rate for all of the genotypes. This was also evidenced in a study conducted on *Aedes albopictus* and *Culex pipiens* in France and on *Culex quinquefasciatus* in the USA, which showed equivalent transmission ratios for G-III and G-V and G-I and G-III strains, respectively [87,88]. Another methodological difference is found in the titers used for blood feeding. Higher titers in the blood meal should make it more likely that the virus will disseminate in the mosquito and, thus, eventually be transmitted. JEV titers in spiked blood used for blood feeding are usually between 10^5^ and 10^7^ PFU/mL [15,84,89]. These high titers are proven realistic as previous studies have shown viraemic reservoir birds (chicks and ducklings) with titers up to 10^6.5^ PFU/mL [90]. Also, temperature conditions can influence the outcome of vector competence studies, as higher temperatures generally increases the competence for flaviviruses [15]. In the competence studies for JEV, the temperatures ranged from 18 to 28 °C. An appropriate temperature should be chosen, one that is relevant to the mosquito population in the area where the study is being conducted. This will be further discussed in the section on vectorial capacity. Finally, the methods used for virus detection (e.g., RT-PCR, virus isolation) can lead to different outcomes in vector competence for the same species. In order to minimize the possible differences in methodology, a standard protocol should be proposed, as suggested for West Nile virus by Vogels et al. [91] and for Zika virus by Azar et al. [92]. In the absence of such a protocol, it is difficult to compare across the different competent species.

Table 4 summarizes the potential and confirmed vectors for JEV. Potential vectors are only proven competent in vector competence experiments, while confirmed vectors are additionally found positive in the field. The following seventeen species can be identified as confirmed vectors: *Aedes albopictus*, *Aedes vexans*, *Aedes vigilax*, *Anopheles tessellatus*, *Armigeres subalbatus*, *Culex annulirostris*, *Culex bitaeniorhynchus*, *Culex fuscocephala*, *Culex gelidus*, *Culex pipiens*, *Culex pipiens pallens*, *Culex pseudovishnui*, *Culex quinquefasciatus*, *Culex sitiens*, *Culex tarsalis*, *Culex tritaeniorhynchus,* and *Culex vishnui.* In addition, the following 10 species are potential vectors: *Aedes detritus*, *Aedes dorsalis*, *Aedes japonicus*, *Aedes kochi*, *Aedes nigromaculis*, *Aedes notoscriptus*, *Culiseta annulata*, *Culiseta incidens*, *Culiseta inornata,* and *Verrallina funerea*. In these, no JEV has been detected in the field to date, which may be due to a lack of surveillance studies.

Based on the extent of their transmission rate, *Armigeres subalbatus*, *Culex annulirostris*, *Culex bitaeniorhynchus*, *Culex gelidus*, *Culex pipiens*, *Culex pseudovishnui*, *Culex quinquefasciatus*, and *Culex tritaeniorhynchus* may be considered the most competent vector species. However, these transmission rates, determined in a particular study, apply to specific mosquito populations tested under certain laboratory conditions and could, therefore, be different in other circumstances.

#### 2.1.3. Vectorial Capacity

Vector competence is only one of the factors that determines whether a specific species will play a role in virus transmission under field conditions. Therefore, the term vectorial capacity was introduced that also takes additional factors, e.g., environmental, behavioral, cellular, and biochemical variables into account [116]. More specifically, vectorial capacity is determined by the density of vectors (abundance) in relation to the host; the probability that the vector feeds on a host; the vector competence; the daily survival rate of a vector; the EIP; and the probability of vectors surviving the EIP [14,82,117]. The EIP is the time interval between the acquisition of the virus and the moment that sufficient virus is present in the saliva to allow further transmission. Vectorial capacity is, therefore, not a single value for a single species, but specific to the vector population at the prevailing climatic conditions in a particular area at a certain moment.

Temperature is one of the most important climatic factors that influences vector capacity, because it has a direct effect on both the daily mosquito survival and the EIP [14], as the proliferation rate of JEV and the metabolism of mosquitoes are affected by temperature. JEV-endemic areas generally have a tropical climate, characterized by warm temperatures and frequent rainfall, and the coolest temperatures are around 20 to 23 °C. As a result, JEV can be transmitted throughout the year in southern tropical areas, although with a higher intensity during the rainy season [3]. When JEV would be introduced in temperate regions where temperatures vary more with the seasons, there would probably not be a year-round JEV transmission. Rather a higher transmission rate would be expected during summer, compared to winter, when few or no vectors are present [118,119]. Low temperatures have been shown to limit the spread of many arboviruses and pose challenges for viruses to overwinter [16]. Nevertheless, several studies have shown that certain mosquitoes, for example *Aedes japonicus*, can transmit JEV vertically to its F1 larvae, providing a potential mechanism of JEV overwintering [98,120].

The abundance of a vector species in a certain region is an important part of the vector capacity calculation. *Culex tritaeniorhynchus* is considered the primary vector for JEV in most endemic areas in Asia, including Japan and Korea [12,121], and *Culex annulirostris* in Australia [122]. However, although *Culex pipiens* is not considered a primary vector, given its high abundance in temperate zones (including Europe) and its competence for JEV, the potential contribution of this vector species to the spread of JEV upon introduction should not be underestimated. In this respect, *Aedes japonicus* also might play a role as it is known to be abundant in certain regions [123,124,125,126,127] and present far beyond its endemic zone (Figure 3). It is one of the world’s most invasive Culicidae species, with a confirmed presence in Europe. While *Aedes japonicus* is a proven vector for JEV [12,97], it has, however, never been found to be positive in the field. For this reason, it is considered a potential secondary vector [12]. In addition, other species, e.g., *Aedes albopictus*, *Aedes dorsalis*, and *Culiseta annulata*, although with lower transmission rates, may contribute to JEV transmission upon introduction [83,87,89,121,128].

Overall vector capacity is the most significant, as well as the most difficult, to calculate. Some of its components are highly variable, e.g., vector–host interactions, vector density, and the probability of daily survival, whereby the latter two can be high in ideal environmental conditions yet decrease very rapidly in the case of unsuitable weather conditions or, for example, human activities involving large-scale vector control measures.

### 2.2. Mosquito Immunity Controlling JEV Replication and Dissemination

Not every infection of a mosquito results in JEV transmission to a new host during a subsequent blood meal. Mechanisms may prevent the development of a virus in a mosquito host that inhibit viral development, dissemination, and transmission. These mechanisms are known as vector immunity.

Key aspects of mosquito immunity include physical barriers, molecular pathways, antimicrobial peptides, and vector microbiome. Over the past thirty years, arbovirus research, focusing mainly on *Aedes* spp. mosquitoes and other flaviviruses, e.g., dengue, West Nile and Zika virus, has identified several mechanisms that limit the replication and dissemination of viruses in mosquitoes [132,133,134,135,136,137].

Recently, comprehensive reviews of the existing knowledge on insect immunity were published [135,137,138,139,140] and we refer readers to those for in depth insights in known molecular mechanisms underlying this immunity. In this review, we provide a summary of the limited existing knowledge on immune mechanisms, which counteract JEV replication in mosquitoes.

#### 2.2.1. Physical and Physiological Barriers

A virus that is ingested through an infectious blood meal must overcome several physical and physiological barriers within a mosquito (Figure 4) before it reaches the saliva and can be successfully transmitted during a subsequent blood meal. These barriers can occur due to genetic (e.g., expression of receptors) or nongenetic determinants (e.g., leaky gut syndrome, i.e., a phenomenon whereby the integrity of the gut wall is compromised) [14,83].

A potential physical mosquito barrier that JEV could encounter is the peritrophic membrane [91]. This membrane forms a physical barrier between the intestinal contents and the epithelia of the midgut. It consists of an extracellular network of chitin, sugars, and proteins. An increase in the thickness of this membrane could, therefore, reduce the chances of a pathogen crossing the intestinal barrier. However, arboviral binding to midgut epithelial cells may occur before the formation of this membrane [141].

There are four main physiological barriers in the mosquito vector, as follows: the midgut infection barrier (1), the midgut escape barrier (2), the salivary gland infection barrier (3), and finally the salivary gland escape barrier (4).

The midgut infection barrier (1) is characterized by the inability of viruses to enter the intestinal cells or to multiply or disseminate to other cells. The midgut escape barrier (2) is the barrier preventing the virus from traversing the basal lamina, that borders the midgut, avoiding the dissemination of the virus throughout the mosquito body. Several mechanisms have been described for how some viruses can cross the basal lamina, as follows: possibly through a “leaky” basal lamina, caused by breakdown and resynthesis after blood feeding, allowing the virus particles to enter the tracheal system and/or hemocoel [142]. This midgut escape barrier has been shown to be temperature dependent for JEV in *Culex pipiens pipiens* [15]. It was demonstrated that at 20 °C JEV was only detected in the epithelial cells in the posterior part of the midgut and in no other tissues, whereas at 25 °C JEV could disseminate to the saliva as JEV RNA was found in the expectorated saliva of 70% of the mosquitoes after 14 days. This indicates that, at 20 °C, the virus was unable to overcome the midgut barrier and consequently could not disseminate to secondary organs, such as the salivary glands. However, it was unclear from these observed results whether the restriction to the midgut was due to lower temperatures that activated antiviral control by the mosquitoes or whether it limited virus replication [15]. It may be that an increase in temperature causes further virus replication, as well as escape from the midgut.

The salivary gland infection barrier (3) is constituted by the basal lamina surrounding the salivary gland, which determines if the virus can disseminate from the midgut and infected fat body via the hemocoel to salivary gland tissue [143]. A study by Takahashi [144] discusses the susceptibility for JEV of each secretory part of salivary glands on transmission efficiency of *Culex tritaeniorhynchus*. They concluded that the salivary gland infection barrier is not a single factor, but that each of its three major secretory parts, i.e., lateral neck cells, lateral acinar cells, and median acinar cells, represent a different level of the barrier. The lateral neck cells are usually the most susceptible and excrete the highest amount of virus in the saliva [144,145].

The salivary gland escape barrier (4) is evidenced by the absence of viral particles in the saliva of infected mosquitoes. This arises from the inability of the viral particles to breach the cell membrane of the salivary gland cells [145]. If a particular virus cannot cross this barrier, no viral particles are found in the mosquito’s saliva, thus preventing transmission. However, if this barrier is crossed, the infected mosquitoes can inoculate virus-infected saliva to a new host during blood feeding.

The analysis of published vector competence studies showed that in four species (*Aedes aegypti, Aedes vigilax, Culex pipiens pallens,* and *Opifex fuscus* [84,93,112]) JEV was only found in the body and legs/wings or optionally the mosquito head, but not in the saliva. A possible explanation is that, in these species, JEV could not cross either the salivary gland infection barrier or the salivary gland escape barrier.

The studies conducted on *Aedes japonicus* [12,97] showed that this species was susceptible to JEV infection. The dissemination rate of the virus was found to be 100% and in 67–100% (depending on genotype used) of these mosquitoes the virus was found in their saliva [12]. This underlines the importance of all of the barriers as a vector competence indicator for this species, since once the midgut is passed and the mosquito is thus “infected”, the virus disseminates “easily” to the salivary glands of the infected mosquitoes, through which it can be transmitted.

#### 2.2.2. Molecular Pathways

RNA interference (RNAi) by small interfering RNA (siRNA) is the central antiviral mechanism in insects, particularly through RNA silencing [137]. This mechanism of small interfering RNA is activated by the binding of dsRNA, which are among others formed during the replication of RNA viruses, to a Dicer-2(dcr2) –R2D2 complex (Figure 5). This complex consists of an RNase III enzyme, which cleaves the dsRNA, and a protein R2D2. The result of this cleavage step is the production of silencing RNAs, which subsequently activate the RNAi pathway upon binding to a multiprotein, the RNA-induced silencing complex. Thereafter, the single-stranded RNA functions as a guide strand to specifically detect and degrade the viral RNA by Argonaute2 (Ago2), a host endonuclease. We only found one study specifically for JEV in relation to this pathway. This study showed that Ago2 suppresses the growth of JEV in the salivary glands of *Aedes aegypti*. RNAi may, therefore, contribute to the low susceptibility of this species for JEV [146].

Besides the small interfering RNA pathway, there are two other known small RNA-based silencing pathways in insects, the microRNA and PIWI-interacting pathways. These all use small RNAs to guide sequence-specific recognition, however, they differ in origin, biogenesis, nature, fate of their targets after recognition, and their biological function [140]. For more detailed explanations of these pathways, we refer the reader to other research [148,149,150].

In addition to RNAi pathways, several other molecular pathways exist that can protect mosquitoes from viral infection, including the Janus kinase-signal transducer and activator of transcription (JAK-STAT), Toll, and immune deficiency pathways (Figure 5). Activation of these initiates the formation of multiprotein complexes consisting of protein kinases, transcription factors, and other regulatory molecules in order to regulate the expression of downstream innate immunity genes, e.g., the genes that encode for antimicrobial peptides (see section below) and the key factors that regulate the innate immune system [137].

The only study that has addressed such pathways in relation to JEV was a study by Lin et al. [151]. In their study, they examined the immune response of mosquitoes to the virus in JEV-infected C6/36 *Aedes albopictus* cells in order to investigate the regulation of the AaSTAT (an *Aedes albopictus* specific cloned mosquito STAT) pathway. Decreased DNA binding activity, as well as decreased tyrosine phosphorylation of AaSTAT, were observed in core extracts from JEV-infected cells, suggesting that JEV infection may disrupt tyrosine phosphorylation of AaSTAT, probably through the induction of cellular phosphatase(s) or the inactivation of JAK or other tyrosine kinase(s) by viral products.

#### 2.2.3. Antimicrobial Peptides

As mentioned above, the formation of a multiprotein complex regulates the activation of downstream signaling and effector responses. This induces the synthesis and secretion of soluble effector molecules, e.g., antimicrobial peptides (AMPs). The AMPs are constitutively released by epithelial cells, such as in the midgut of mosquitoes, where they prevent overgrowth of the gut microbiota, thus, playing an important role in tuning the immune response by tolerating symbiosis and controlling microbial growth [152]. The AMPs in mosquitoes are primarily regulated by the Imd pathway [153].

Recent studies have shown that the AMP defensin, which is one of the crucial immune effectors in insects [154], plays an important role in facilitating JEV infection and potential transmission in mosquitoes. An initial study by Liu et al. [155] showed that mosquito defensins (*Culex pipiens pallens* defensin A and *Aedes albopictus* defensin C) facilitate the adsorption of JEV to target cells by binding to a specific part of the viral envelope protein of JEV. Moreover, under natural conditions, the local infection of the midgut leads to rapid upregulation and extracellular secretion of defensins [156]. In a subsequent study, the same group showed that defensin regulates cell-surface proteins [157]. A potential antiviral cell-surface protein (HSC70B) was significantly downregulated by both JEV infection and by defensin treatment. This protein inhibits JEV adsorption, indicating that mosquito defensin indirectly affects JEV adsorption by regulating cell-surface antiviral protein expression. Together, these two studies show that defensins have a (in)direct effect on both JEV infection and transmission.

#### 2.2.4. Vector Microbiome

The microbiome of insects is composed of bacteria, fungi, viruses, and helminths and has the ability to reduce the vector competence for arboviruses and other pathogens. This reduction can occur through different mechanisms, e.g., the activation of the immune response, competition for resources, changing the physical status, or the production of antiviral molecules [152,158]. These symbiotic microorganisms reside in the gut, lumen and/or hemocoel of arthropod vectors [152]. In the context of vector immunity, the gut is of particular importance because it is the first and most extensive area exposed to pathogens [159]. There is a known high diversity in the composition of the microbial community in the midgut as they are frequently acquired from the habitats and are, thus, shaped by the environmental conditions [152]. As mentioned in the previous section, symbiosis of the microbiota is regulated by AMPs. Furthermore, reactive oxygen species play a key role in the regulation of vector microbiota homeostasis.

The gut microbiome is also involved in the formation of the peritrophic membrane [160], one of the physical barriers between the intestinal contents and the epithelia of the midgut, as discussed earlier in the section on physical and physiological barriers.

*Wolbachia* is the most extensively studied bacteria of the mosquito microbiome. In *Aedes aegypti*, *Wolbachia* infection has been found to increase the resistance to RNA virus infection. The molecular mechanisms involved in its protection are, however, not yet fully understood [161]. In contrast, in *Armigeres subalbatus,* no significant difference was shown between *Wolbachia*-infected and -free colonies. In their study, it is suggested that *Wolbachia* does not play a role in the resistance of salivary gland cells to JEV infection. Therefore, it is probable that the salivary gland escape barrier is not impaired by *Wolbachia* infection in this species [162].

The microbiome seems to specifically influence vector competence for JEV in *Culex bitaeniorhynchus*, since Mourya and Soman [163] showed that tetracycline treatment of this species increased their infection rate. Namely, twice as many (i.e., 43.41%) of the antibiotic-treated mosquitoes were positive for JEV after an infected bloodmeal, compared to untreated mosquitoes (22.5%). Similar observations have already been made in several other studies focusing on other arboviruses [161,164,165,166,167,168].

## 3. Conclusions

In this review, the current knowledge on the vector competence and vector capacity of mosquitoes for JEV is presented, as well as the limited knowledge on the underlying mechanisms that influence these parameters, e.g., vector immunity, abundance, and the effects of climate change.

Regarding vector competence, differences in methodology make it difficult to compare studies and draw definitive conclusions on which species are more competent than others, as their transmission rates may differ due to a difference in methodology. Results from vector-competence studies, combined with field-detection studies, indicate that 17 species are important to take into account. These all have the potential to transmit JEV and have already been found to be positive in the field, which makes them currently known vectors for JEV. Among these, *Culex tritaeniorhynchus* and *Culex annulirostris* are considered primary vectors in endemic areas. Additionally, *Culex pipiens,* and potentially *Aedes japonicus,* could be considered as important vectors in the case of the introduction of JEV into new areas.

The information gathered on vector immunity provides an indication of the underlying mechanisms that determine vector competence. However, very little is known about the barriers and conditions for the replication and transmission of JEV at the mosquito species level. A better understanding of the immunity, physiology, genetics, and microbiome of mosquito vectors in relation to JEV will be required in order to identify novel innovative vector control strategies that could help in reducing JEV transmission. We therefore advocate to invest in such studies.

## 4. Methods

A PubMed database search (on 14 December 2021) using the query term “Japanese encephalitis virus” yielded 5027 articles. Based on the title we retained all articles which could contain pertinent information on JEV–mosquito interactions (Figure 6). From this, an initial selection was made by excluding articles on diagnostic methods, vaccine production or vaccination studies, virus propagation techniques, case studies, epidemiological studies, and articles on the immunological relationship of JEV with other viruses. This resulted in 193 potentially relevant articles, which we screened for relevance by reading the abstracts, after which we excluded all articles that addressed biocontrol strategies, surveillance studies without species specification, insect-specific flaviviruses, and ecological studies. This resulted in a total of 114 manuscripts specifically dealing with JEV–vector interactions, from which we then extracted the data reported in this review. For some articles [20,23,29,30,37,50,52,57,58,59,61,65,68,99,100,102,103,104,105,106,107,111,113,114] the full text was not available, for these the information in the tables was taken from the abstracts.

## Figures and Tables

**Figure 1 pathogens-11-00317-f001:**
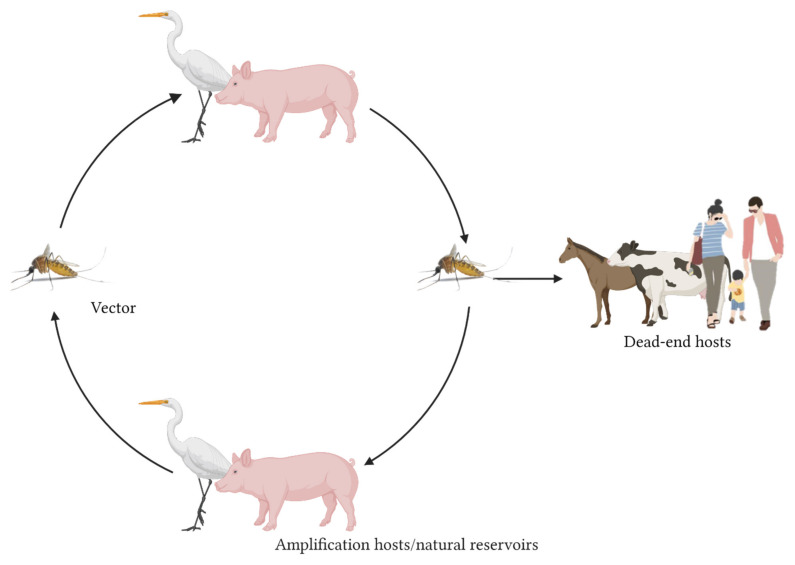
Transmission cycle of JEV. Competent mosquitoes transmit JEV between natural reservoirs, e.g., Ardeid birds and amplifying hosts, e.g., pigs. Horses, cattle, and humans are considered dead-end hosts. Created with BioRender.com.

**Figure 2 pathogens-11-00317-f002:**
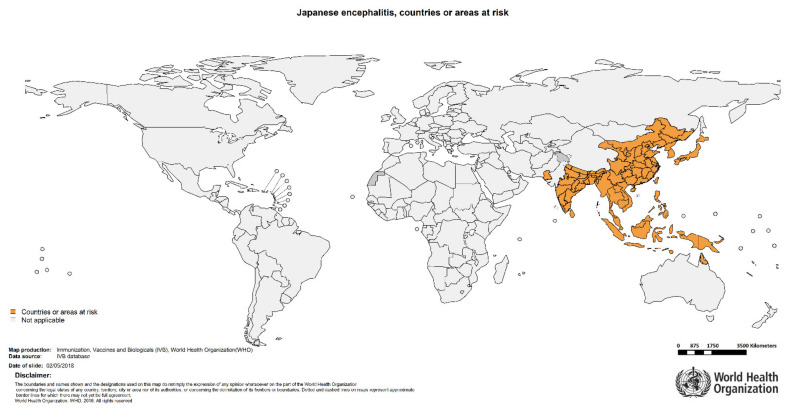
Map issued by the WHO showing the current countries or areas at risk for JEV [3]. Reprinted with permission from BioRender.com.

**Figure 3 pathogens-11-00317-f003:**
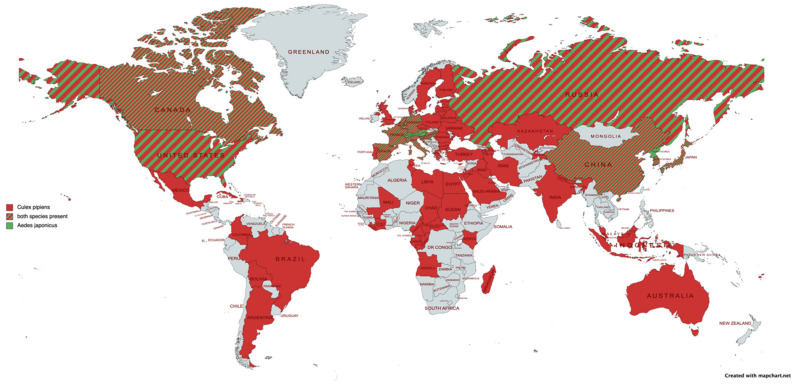
Map showing the worldwide distribution of *Aedes japonicus* (green) and *Culex pipiens* (red). This map was created based on a study by Peach et al. [129], the Invasive Species Compendium of CABI [130], and the ECDC mosquito maps [131].

**Figure 4 pathogens-11-00317-f004:**
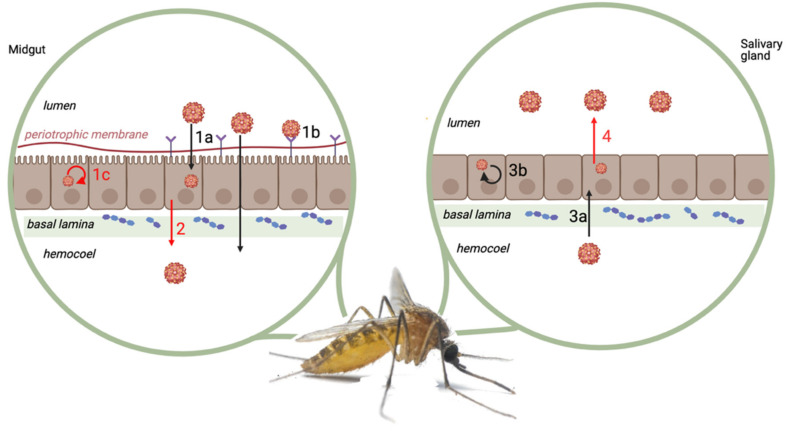
Four major mosquito barriers. (1) Midgut infection barrier, which results from either the inability of the virus to enter the midgut cells (1a), the absence of suitable receptors (1b), and/or the inability of the virus to replicate within the midgut cells (1c). (2) Midgut escape barriers. (3) Salivary gland infection barrier, which can result from either the ability of the virus to enter the salivary gland cells (3a) and/or the ability of the virus to replicate within the salivary gland cells (3b). (4) Salivary gland escape barrier. Barriers for which JEV specific information exist are shown in red. Adapted from Vogels et al., 2017 [91]. Created with BioRender.com.

**Figure 5 pathogens-11-00317-f005:**
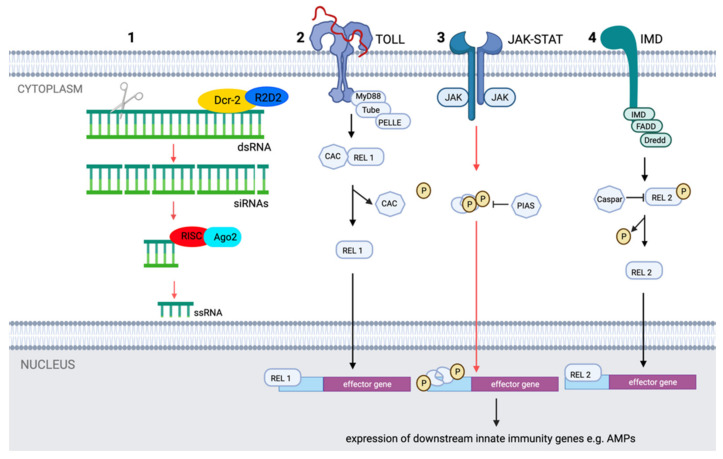
The major Culicidae innate immune pathways RNAi (1), Toll (2), JAK-STAT (3), and immune deficiency pathways (IMD) (4). All the names of the genes shown correspond to the nomenclature adapted from Terradas et al., (2017) [147] and Lee et al., (2019) [137]. Molecular pathways for which JEV specific information exist are shown in red. Created with BioRender.com.

**Figure 6 pathogens-11-00317-f006:**
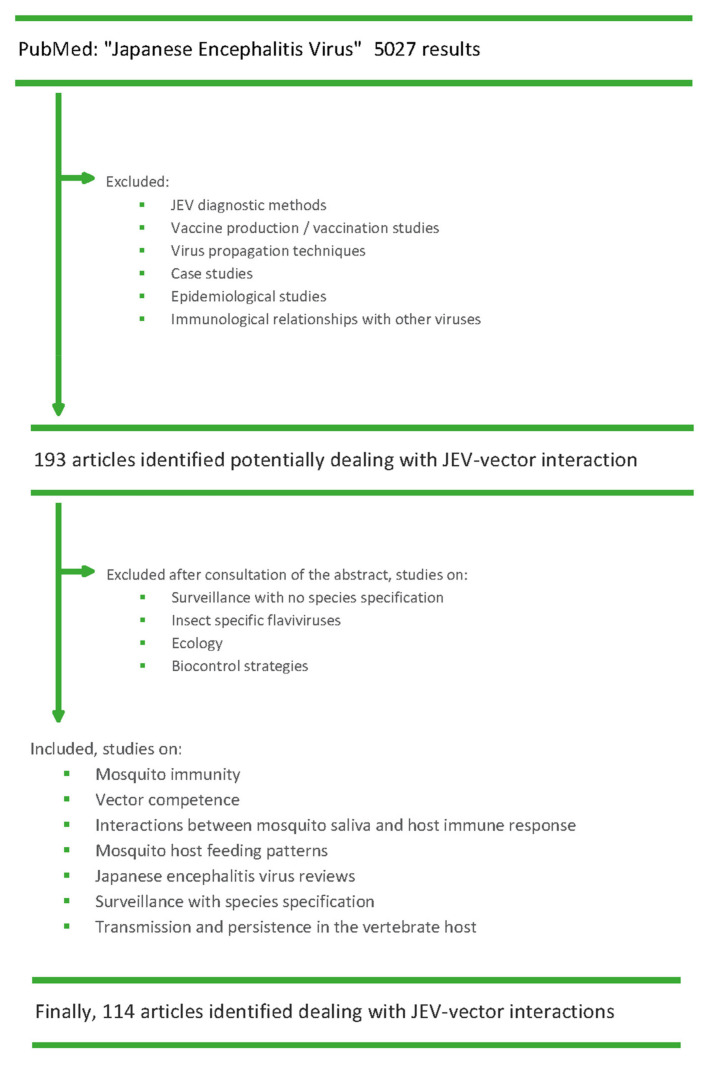
Flowchart of the articles identified and screened for this review.

**Table 1 pathogens-11-00317-t001:** Overview of field-collected mosquitoes in which JEV was detected. Underlined species have been proven to be competent vectors through competence studies (see Table 3).

Mosquito Species	Country of Sampling	JEV Genotype and/or Strain	Detection Method	Total # Tested	# JEV Positive Pools/Total # of Pools	MIR (# Positive Pools/Total Tested) × 100	Reference
*Aedes* *albopictus* (Skuse, 1895)	Taiwan	G-I and III	RT-PCR	177	1/25	0.56%	[19]
	Taiwan	ND	Isolation	ND	20	ND	[20]
*Aedes butleri* (Theobald, 1901)	Malaysia	ND	Isolation and RT-PCR	3950	4/79	0.1%	[21,22]
*Aedes curtipes* (Edwards, 1915)	Malaysia	ND	ND	ND	ND	ND	[23]
*Aedes lineatopennis* (Ludlow, 1905)	Malaysia	ND	Isolation and RT-PCR	300	1/6	0.33%	[22]
*Aedes vexans* (Meigen, 1830)	Taiwan	G-I and III	RT-PCR	246	3/32	1.22%	[19]
	Taiwan	ND	Isolation	ND	1	ND	[20]
	Taiwan	ND	RT-PCR	61	1/9	1.64%	[24]
*Aedes vigilax* (Skuse, 1889)	Australia	ND	Isolation	ND	1	ND	[25]
	Australia	G-II	Isolation	3073	1	0.03%	[26]
*Anopheles annularis* (Wulp, 1884)	Indonesia	ND	Isolation	250	1/28	0.4%	[27]
*Anopheles barbirostris* (Wulp, 1884)	India	ND	ELISA and *Toxo-*IFA	22	1/8	4.55%	[28]
	India	ND	ND	ND	ND	ND	[29]
*Anopheles minimus* (Theobald, 1901)	Taiwan	G-I and III	RT-PCR	18	1/7	5.56%	[19]
*Anopheles hyrcanus* (Pallas, 1771)	India	ND	ND	ND	ND	ND	[30]
	India	ND	ND	ND	ND	ND	[29]
*Anopheles pallidus* (Theobald, 1901)	India	ND	ELISA and *Toxo-*IFA	28	1/12	3.57%	[28]
*Anopheles peditaeniatus* (Leicester, 1908)	India	ND	Isolation	6306	1/133	0.02%	[31]
*Anopheles sinensis* (Wiedemann, 1828)	China	G-III	RT-PCR	ND	12	ND	[32]
	China	ND	RT-PCR	2802	5/55	0.18%	[33]
	China	G-I	Isolation	14,170	3	0.02%	[34]
	Taiwan	G-I and III	RT-PCR	2638	6/119	0.23%	[19]
*Anopheles subpictus* (Grassi, 1899)	India	ND	ELISA and *Toxo-*IFA	1432	7/67	0.49%	[28]
	India	ND	ELISA and *Toxo-*IFA	ND	ND	ND	[35]
	India	ND	ELISA and *Toxo-*IFA	6550	4/131	0.06%	[36]
	India	ND	ND	ND	ND	ND	[37]
*Anopheles tessellatus* (Theobald, 1901)	Taiwan	G-I and III	RT-PCR	536	2/31	0.37%	[19]
*Anopheles vagus* (Dönitz, 1902)	Indonesia	ND	Isolation	2700	1/42	0.04%	[27]
*Armigeres subalbatus* (Coquillet, 1898)	China	G-I	Isolation	394	2	0.51%	[34]
	China	G-III	RT-PCR	ND	3	ND	[32]
	India	ND	ELISA and *Toxo*-IFA	110	1/21	0.91%	[28]
	Taiwan	G-I and III	RT-PCR	225	3/30	1.33%	[19]
	Taiwan	ND	Isolation	ND	8	ND	[20]
*Coquillettidia ochracea* (Theobald, 1903)	China	ND	RT-PCR	155	1/6	0.65%	[38]
*Culex annulirostris* (Skuse, 1889)	Australia	G-III	RT-PCR	2871	8/134	0.28%	[39]
	Australia	ND	Isolation	23,890	42	0.18%	[25]
*Culex annulus* (Theobald, 1901)	Taiwan	G-I and III	RT-PCR	991	9/79	0.91%	[19]
	Taiwan	ND	Intracerebral inoculation of mice	1338	3	0.23%	[40]
	Taiwan	ND	Isolation	54,910	31/703	0.06%	[41]
	Taiwan	ND	Isolation	ND	7/31	ND	[42]
	Taiwan	ND	Isolation	ND	1	ND	[20]
*Culex bitaeniorhynchus* (Giles, 1901)	India	ND	ELISA and *Toxo-*IFA	44	1/9	2.28%	[28]
	Korea	G-I	RT-PCR	344	1/26	0.29%	[43]
	Korea	G-I and V	RT-PCR	1960	2/175	0.1%	[44]
	Korea	ND	RT-PCR	1140	1/45	0.09%	[45]
	Malaysia	ND	Isolation and RT-PCR	550	3/11	0.55%	[21]
*Culex epidesmus* (Theobald, 1905)	India	ND	Isolation	ND	1	ND	[30]
*Culex fuscanus* (Wiedemann, 1820)	Taiwan	ND	Isolation	ND	1	ND	[20]
*Culex fuscocephala* (Theobald, 1907)	India	ND	Isolation	14,664	1/257	0.007%	[31]
	India	ND	Isolation	15,250	6/305	0.04%	[46]
	Indonesia	ND	Isolation	ND	1	ND	[47]
	Malaysia	ND	Isolation and RT-PCR	3800	2/76	0.05%	[22]
	Taiwan	G-I and III	RT-PCR	394	3/19	0.76%	[19]
	Taiwan	ND	RT-PCR	1150	1/23	0.09%	[48]
	Taiwan	ND	Isolation	22,895	19/282	0.08%	[49]
	Thailand	ND	Isolation or HI and CF assays	142,375	2	0.001%	[50]
	Thailand	ND	ND	ND	2	ND	[50]
*Culex gelidus* (Theobald, 1901)	Australia	ND	RT-PCR	4872	3	0.06%	[51]
	India	ND	Isolation	6038	3/127	0.05%	[31]
	India	ND	ND	ND	ND	ND	[52]
	India	ND	ELISA and *Toxo-*IFA	7485	4/177	0.05%	[53]
	India	ND	Isolation	9700	5/194	0.05%	[46]
	India	ND	ELISA and *Toxo-*IFA	8750	17/175	0.2%	[54]
	India	ND	ELISA and *Toxo-*IFA	17,678	12/403	0.07%	[55]
	India	ND	ELISA and *Toxo-*IFA	16,658	10	0.06%	[56]
	Indonesia	ND	ND	ND	1	ND	[57]
	Indonesia	ND	Isolation	7144	2/154	0.03%	[47]
	Malaysia	ND	Isolation and RT-PCR	11,200	12/224	0.11%	[22]
	Malaysia	ND	ND	ND	1	ND	[58]
	Malaysia	ND	ND	ND	ND	ND	[23]
	Sri Lanka	ND	Isolation	13,043	4	0.03%	[59]
	Thailand	ND	Isolation or HI and CF assays	11,495	3	0.03%	[50]
	Thailand	Not given	Inoculation in mice	3097	18	0.6%	[60]
	Vietnam	Not given	ND	ND	ND	ND	[61]
*Culex infula* (Theobald, 1901)	India	ND	ELISA and *Toxo-*IFA	119	2/16	1.68%	[28]
*Culex orientalis* (Edwards, 1921)	Korea	G-V	RT-PCR	498	5/83	1%	[62]
*Culex pipiens* (Linnaeus, 1758)	Italy	G-III	RT-PCR	ND	1/57	ND	[63]
	Korea	G-I	RT-PCR	736	4/64	0.54%	[43]
	Korea	G-I	RT-PCR	11,237	4/804	0.04%	[44]
	Korea	G-V	RT-PCR	9295	1/264	0.01%	[62]
	China	G-I	RT-PCR	1540	1/256	0.06%	[64]
*Culex pipiens pallens* (Coquillett, 1898)	China	ND	RT-PCR	6465	10/132	0.15%	[38]
*Culex pseudovishnui* (Colless, 1957)	India	ND	ND	ND	1	ND	[65]
	India	ND	ELISA and *Toxo-*IFA or RT-PCR	ND	3/107	ND	[66]
	India	ND	ELISA and *Toxo-*IFA or RT-PCR	1406	1	0.07%	[67]
	India	ND	ND	ND	ND	ND	[68]
	India	ND	ND	ND	ND	ND	[37]
*Culex quinquefasciatus* (Say, 1823)	India	ND	ELISA and *Toxo-*IFA	59	1/13	1.69%	[28]
	India	ND	Isolation	304	1/18	0.33%	[31]
	Malaysia	ND	Isolation and RT-PCR	2400	1/48	0.4%	[22]
	Taiwan	G-I and III	RT-PCR	1333	2/74	0.15%	[19]
	Taiwan	ND	Isolation	ND	7	ND	[20]
	Thailand	ND	Isolation	1023	2/25	0.2%	[69]
	Vietnam	G-III	RT-PCR	ND	30	ND	[70]
*Culex rubithoracis* (Leicester, 1908)	Taiwan	ND	RT-PCR	130	4/22	3.08%	[24]
*Culex sitiens* (Wiedemann, 1828)	Australia	ND	RT-PCR	18,680	5	0.03%	[51]
	Australia	ND	RT-PCR	22,833	1	0.004%	[71]
	Australia	G-II	Isolation	25,292	42	0.16%	[26]
	Australia	G-I	Isolation	44,755	1	0.002%	[72]
	Malaysia	ND	Isolation and RT-PCR	400	2/8	0.5%	[21]
	Papua New Guinea	G-II	Isolation	245,483	3	0.001%	[73]
	Taiwan	ND	RT-PCR	604	1/34	0.17%	[24]
	Taiwan	ND	Isolation	ND	2	ND	[20]
	Vietnam	G-I and III	RT-PCR	ND	73	ND	[70]
*Culex tarsalis* (Coquillett, 1896)	China	G-III	RT-PCR	ND	57	ND	[32]
*Culex tritaeniorhynchus* (Giles, 1901)	Cambodia	G-I	Isolation	7218	1/729	0.01%	[74]
	China	ND	RT-PCR	6610	31/135	0.47%	[38]
	China	ND	RT-PCR	15,795	24/158	0.15%	[33]
	China	G-I	Isolation	37,119	15	0.04%	[34]
	China	G-I	RT-PCR	3945	4/255	0.1%	[64]
	China	G-I	RT-PCR	6490	15/149	0.23%	[75]
	China	G-I	RT-PCR	2927	3/152	0.1%	[76]
	India	ND	ELISA and *Toxo-*IFA	9937	10/245	0.10%	[28]
	India	ND	Isolation	12,161	2/272	0.02%	[31]
	India	ND	Isolation	206,424	58/4128	0.03%	[46]
	India	ND	ELISA and *Toxo-*IFA	ND	ND	ND	[35]
	India	ND	ELISA and *Toxo-*IFA	7485	4/177	0.05%	[53]
	India	ND	ELISA and *Toxo-*IFA	45,100	62/902	0.14%	[54]
	India	ND	ELISA and *Toxo-*IFA	21,005	13/429	0.06%	[53]
	India	ND	ELISA and *Toxo-*IFA	14,358	14/309	0.1%	[55]
	India	ND	ELISA and *Toxo-*IFA	100,611	64	0.06%	[56]
	India	ND	ELISA and *Toxo-*IFA or RT-PCR	862	2	0.23%	[67]
	Indonesia	ND	Isolation	112,398	1/596	0.0009%	[27]
	Indonesia	ND	Isolation	18,486	19/359	0.1%	[47]
	Japan	G-I	Isolation	3328	3/141	0.09%	[77]
	Korea	G-I	RT-PCR	2880	29/121	1.01%	[43]
	Korea	G-I and V	RT-PCR	55,135	92/2031	0.17%	[44]
	Korea	ND	RT-PCR	5909	50/207	0.85%	[45]
	Malaysia	ND	Isolation and RT-PCR	1300	3/26	0.23%	[21]
	Malaysia	ND	Isolation and RT-PCR	36,550	24/731	0.07%	[22]
	Singapore	G-II	RT-PCR	882	5/88	0.57%	[78]
	Sri Lanka	ND	Isolation	17,436	4	0.02%	[59]
	Taiwan	ND	Isolation	16,776	18/267	0.11%	[41]
	Taiwan	ND	RT-PCR	28,773	95/1061	0.33%	[24]
	Taiwan	ND	RT-PCR	37,500	25/750	0.07%	[48]
	Taiwan	ND	Isolation	ND	97	ND	[20]
	Taiwan	G-I and III	RT-PCR	89,189	468/2242	0.52%	[19]
	Taiwan	ND	Isolation	ND	2/6	ND	[42]
	Thailand	ND	Isolation or HI and CF assays	183,140	8	0.004%	[50]
	Thailand	ND	Isolation	290,126	34	0.01%	[79]
	Vietnam	G-I and III	RT-PCR	ND	3	ND	[70]
	Vietnam	G-I	Isolation	4199	3/131	0.07%	[80]
*Culex vishnui* (Theobald, 1901)	India	ND	ELISA and *Toxo-*IFA or RT-PCR	1512	3	0.2%	[67]
	India	ND	ND	ND	ND	ND	[37]
	India	ND	ELISA and *Toxo-*IFA	2787	1/61	0.04%	[53]
	India	ND	Isolation	54,007	22/1080	0.04%	[46]
	Indonesia	ND	Isolation	ND	1	ND	[47]
	Malaysia	ND	Isolation and RT-PCR	1650	4/33	0.24%	[21]
	Thailand	ND	Isolation	8408	1	0.01%	[79]
	Vietnam	G-I	Isolation	1542	2/46	0.13%	[80]
*Culex whitmorei* (Giles, 1904)	India	ND	ELISA and *Toxo-*IFA	47	2/17	4.26%	[28]
	Sri Lanka	ND	Isolation	167	1	0.6%	[59]
*Mansonia bonneae/dives* (Edwards, 1930/Schiner, 1868)	Malaysia	ND	ND	ND	ND	ND	[23]
*Mansonia annulifera* (Theobald, 1901)	India	ND	ELISA and *Toxo-*IFA	ND	ND	ND	[35]
	India	ND	ELISA and *Toxo-*IFA	4530	3	0.07%	[56]
*Mansonia indiana* (Edwards, 1930)	India	ND	ELISA and *Toxo-*IFA	12,362	12	0.1%	[56]
	India	ND	ELISA and *Toxo-*IFA	62	2/13	3.23%	[28]
*Mansonia uniformis* (Theobald, 1901)	India	ND	ELISA and *Toxo-*IFA	ND	ND	ND	[35]
	India	ND	ELISA and *Toxo-*IFA	14,503	5	0.03%	[56]
	Malaysia	ND	ND	ND	ND	ND	[23]
	Sri Lanka	ND	ND	ND	ND	ND	[59]
	Taiwan	G-I and III	RT-PCR	75	1/19	1.33%	[19]

RT-PCR = reversed transcription polymerase chain reaction; HI = hemagglutination inhibition; CF = complement fixation; *Toxo*-IFA = indirect immunofluorescence assay on inoculated *Toxorhynchites splendens* mosquito larvae.

**Table 2 pathogens-11-00317-t002:** Overview of mosquito screening studies in which JEV was not detected.

Mosquito Species	Country of Sampling	JEV Genotype and/or Strain	Detection Method	Total # Tested	# JEV Positive Pools	Reference
*Aedes aegypti* (Linnaeus, 1762)	Taiwan	G-I and III	RT-PCR	3	0/2 pools	[19]
*Aedes albolateralis* (Theobald, 1908)	Taiwan	G-I and III	RT-PCR	1	0/1 pools	[19]
*Aedes albopictus **	Korea	G-V	RT-PCR	564	0/64 pools	[62]
	Korea	G-I	RT-PCR	66	0/15 pools	[43]
*Aedes dorsalis* (Meigen, 1830)	Korea	G-V	RT-PCR	6	0/6 pools	[62]
*Aedes koreicus* (Edwards, 1917)	Korea	G-I	RT-PCR	181	0/24 pools	[43]
*Aedes lineatopennis **	Korea	G-I	RT-PCR	1	0/1 pools	[43]
	Thailand	ND	Isolation or HI and CF assays	16,230	0 pools	[50]
*Aedes mediolineatus* (Theobald, 1901)	Thailand	ND	Isolation or HI and CF assays	15,122	0 pools	[50]
*Aedes nipponicus* (LaCasse & Yamaguti, 1948)	Korea	G-I	RT-PCR	1	0/1 pools	[43]
*Aedes penghuensis* (Lien, 1968)	Taiwan	G-I and III	RT-PCR	283	0/10 pools	[19]
*Aedes togoi* (Theobald, 1907)	Taiwan	G-I and III	RT-PCR	1	0/1 pools	[19]
*Aedes vexans **[81]	Thailand	ND	Isolation or HI and CF assays	11,022	0 pools	[50]
*Aedes vexans nipponii* (Theobald, 1907)	Korea	G-I	RT-PCR	2091	0/106 pools	[43]
*Anopheles ludlowae* (Theobald, 1903)	Taiwan	G-I and III	RT-PCR	1	0/1 pools	[19]
*Armigeres subalbatus **	Korea	G-V	RT-PCR	1132	0/145 pools	[62]
	Korea	G-I	RT-PCR	23	0/9 pools	[43]
*Coquillettidia crassipes* (Van der Wulp, 1881)	Taiwan	G-I and III	RT-PCR	47	0/3 pools	[19]
*Coquillettidia ochracea **	Korea	G-V	RT-PCR	115	0/14 pools	[62]
*Culex bitaeniorhynchus **	Korea	G-V	RT-PCR	50	0/16 pools	[62]
	Taiwan	G-I and III	RT-PCR	60	0/7 pools	[19]
*Culex brevipalpis* (Giles, 1902)	Taiwan	G-I and III	RT-PCR	1	0/1 pools	[19]
*Culex fuscanus **	Taiwan	G-I and III	RT-PCR	4	0/3 pools	[19]
*Culex fuscocephalus**	Thailand	ND	Isolation	9140	0 pools	[79]
*Culex gelidus **	Thailand	ND	Isolation	17,530	0 pools	[79]
*Culex hayshii* (Yamada, 1917)	Korea	G-V	RT-PCR	4	0/2 pools	[62]
*Culex inatomii* (Kaminura & Wada, 1974)	Korea	G-V	RT-PCR	470	0/16 pools	[62]
	Korea	G-I	RT-PCR	1	0/1 pools	[43]
*Culex mimeticus* (Noè, 1899)	Korea	G-V	RT-PCR	1	0/1 pools	[62]
	Taiwan	G-I and III	RT-PCR	1	0/1 pools	[19]
*Culex murrelli* (Lien, 1968)	Taiwan	G-I and III	RT-PCR	39	0/3 pools	[19]
*Culex nigropunctatus* (Edwards, 1926)	Taiwan	G-I and III	RT-PCR	9	0/1 pools	[19]
*Culex orientalis **	Korea	G-I	RT-PCR	3	0/2 pools	[43]
*Culex quinquefasciatus **	Thailand	ND	Isolation	73	0 pools	[79]
*Culex rubensis* (Sasa & Takahashi, 1948)	Korea	G-V	RT-PCR	1	0/1 pools	[62]
*Culex rubithoracis **	Taiwan	G-I and III	RT-PCR	65	0/8 pools	[19]
*Culex sitiens **	Taiwan	G-I and III	RT-PCR	6295	0/128 pools	[19]
*Culex tritaeniorhynchus **	Korea	G-V	RT-PCR	10	0/7 pools	[62]
*Culex vagans* (Wiedemann, 1828)	Korea	G-V	RT-PCR	5	0/2 pools	[62]
*Culex vishnui **	Thailand	ND	Isolation or HI and CF assays	22,005	0 pools	[50]
*Culex whitmorei **	Thailand	ND	Isolation	530	0 pools	[79]
*Culiseta bergrothi* (Edwards, 1921)	Korea	G-V	RT-PCR	1	0/1 pools	[62]
*Mansonia uniformis **	Korea	G-V	RT-PCR	2176	0/66 pools	[62]
	Korea	G-I	RT-PCR	1	0/1 pools	[43]
*Tripteroides bambusa* (Yamada, 1917)	Korea	G-V	RT-PCR	30	0/9 pools	[62]
*Uranotaenia macfarlanei* (Edwards, 1914)	Taiwan	G-I and III	RT-PCR	1	0/1 pools	[19]

RT-PCR = reversed transcription polymerase chain reaction; HI = hemagglutination inhibition; CF = complement fixation * These species have been detected positive in other studies/regions.

**Table 3 pathogens-11-00317-t003:** Detailed overview of vector competence studies in different mosquito species for JEV. Underlined species have been detected positive in the field (Table 1).

Mosquito Species	Origin of Mosquito Colony	JEV Strain Used	Cell Type Used for Virus Production	Virus Titer in Bloodmeal	Blood Origin	Feeding Method	Inc. Temperature	Inc. Period (Days)	# Mosquitoes	% Infected *	% Disseminated **	% Transmission Competent ***	Detection Method	Reference
* Aedes aegypti *	Australia, Townsville	G-II (TS3306)	C6/36 and porcine stable-equine kidney cells	10^4.5±0.1^ CCID_50_/mL	Heparinized rabbit	Glass membrane feeder with pig intestine	28 °C	14–15	60	27%	17%	ND	Porcine stable-equine kidney cells	[93]
* Aedes * * albopictus *	Australia, Masig Island	G-I (TS00)	Porcine stable equine kidney and C6/36 cells	3.5 ± 0.3 log_10_ CCID_50_/mL	Washed defibrinated sheep	Cotton pledged	28 °C	14	25	20%	16%	16%	Vero cells	[94]
	France, Montpellier and Nice	G-III (RP-9) and G-V (XZ0934)	Chicken fibroblast-derived DF1 cells	8 × 10^6^ FFU/mL	Washed rabbit erythrocytes	Cotton pledgets	26 °C	7–13	5–20	70–100%	57–100% ^◊^	20–63%	BHK-21 cells	[87]
	Taiwan, Tapei and Taichung County	ND (Sanshia MQ1-2)	C6/36 cells	5.42 log_10_ WMICLD5_0_	NA	Intraperitoneal inoculated mice	26–28 °C	14	20	ND	ND	27–45%	BHK-21 cells	[95]
*Aedes detritus* (Haliday, 1833)	UK, Northwest England	G-V (Muar)	Vero cells	4 log_10_ PFU/mL	Defibrinated horse	Hemotek with Parafilm membrane	23 and 28 °C	0–21	6–32	32–100%	20–100%	3–67%	Vero cells	[85]
* Aedes dorsalis *	US	G-III (Nakayama)	ND	ND	Defibrinated rabbit	Cotton pledgets	27 °C	16	2–10	ND	ND	4% ^#^	Development of encephalitis in laboratory-reared mice	[96]
*Aedes* *japonicus* (Theobald, 1901)	Germany, Stuttgart	ND	ND	ND	Human	Cotton pledgets	25 °C	0–14	3–4	100%	ND	ND	RT-qPCR	[97]
	Japan, Narita	G-I (17CxIT-I4-D31), 3 (JaGAr 01) and V (Muar)	C6/36 cells	8.9, 8.6, and 7.1 log_10_ FFU/mL	Defibrinated rabbit	Hemotek with pig intestine membrane	27 °C	7–14	3–36	2–19%	2–19%	2–16%	RT-qPCR or FFA in Vero cells	[12]
	Japan, Sapporo	G-III (JANAr-5681)	C6/36 cells	6.2 PFU/mL (blood) and 3.7 PFU/mL (chicken)	ND	Cotton pledgets or viremic chicken	20 or 28 °C	0–20	40	67.5%	ND	50%	BHK-21 cells and IFA	[98]
*Aedes kochi* (Dönitz, 1901)	Australia, Bamaga and Cairns (wild)	G-II (TS3306)	C6/36 and porcine stable-equine kidney cells	10^4.5±0.1^ CCID_50_/mL	Heparinized rabbit	Glass membrane feeder with pig intestine membrane	28 °C	14–15	37	19%	ND	6%	Detection of virus in brain aspirates of recipient suckling mice	[93]
*Aedes* *nigromaculis* (Ludlow, 1906)	US	G-III (Nakayama)	ND	ND	Defibrinated rabbit	Cotton pledgets	27 °C	16	11–100	ND	ND	4% ^#^	Development of encephalitis in laboratory-reared mice	[96]
*Aedes* *notoscriptus*(Skuse, 1889)	Australia, Closeburn	G-II (TS3306)	C6/36 and porcine stable-equine kidney cells	10^4.5±0.1^ CCID_50_/mL	Heparinized rabbit	Glass membrane feeder with pig intestine	28 °C	13/14	11–48	27%	8%	27%	Porcine stable-equine kidney cells	[93]
* Aedes vexans *	Guam	ND (Okinawa, human 1945)	ND	ND	NA	Inoculated mice	ND	ND	ND	ND	ND	Successful	Development of encephalitis in laboratory-reared mice	[99]
* Aedes vexans nipponii *	Japan, Sapporo	G-III (JANAr-5681)	C6/36 cells	6.2 PFU/mL (blood) and 3.7 PFU/mL (chicken)	ND	Cotton pledgets or viremic chicken	20 or 28 °C	0–20	12	25%	ND	ND	BHK-21 cells and IFA	[98]
* Aedes vigilax *	Australia, Cairns (wild)	G-II (TS3306)	C6/36 and porcine stable-equine kidney cells	10^4.5±0.1^ CCID_50_/mL	Heparinized rabbit	Glass membrane feeder with pig intestine membrane	28 °C	14–15	75	57%	ND	17%	Detection of virus in brain aspirates of recipient suckling mice	[93]
	Australia, Redlands Shire	G-II (TS3306)	C6/36 and porcine stable-equine kidney cells	10^7.1±0.1^ CCID_50_/mL	Heparinized rabbit	Glass membrane feeder with pig intestine	28 °C	9–13	4–62	19–39%	18–39%	0%	Porcine stable-equine kidney cells	[93]
* Anopheles * * tessellatus *	India	G-I (733913)	NA	ND	Viremic chickens	NA	ND	11	13	ND	ND	31%	Transmission to chickens	[100]
* Armigeres subalbatus *	Taiwan, Liu-Chiu	G-III (T1P1)	C6/36	1.25 × 10^7^ PFU/mL	Rabbit	Drop of blood	ND	1–20	8–14	ND	ND	0–79%	IFAT	[101]
* Culex * * annulirostris *	Guam	ND (Okinawa, human 1945)	ND	ND	NA	Inoculated mice	ND	ND	ND	ND	ND	Successful	encephalitis in laboratory-reared mice	[99]
	Australia, Bamaga and Cairns (wild)	G-II (TS3306)	C6/36 and porcine stable-equine kidney cells	10^4.5±0.1^ CCID_50_/mL	Heparinized rabbit	Glass membrane feeder with pig intestine membrane	28 °C	14–15	25–57	93%	ND	56%	Detection of virus in brain aspirates of recipient suckling mice	[93]
	Australia, Brisbane	G-II (TS3306)	C6/36 and porcine stable-equine kidney cells	10^4.5±0.1^ CCID_50_/mL	Heparinized rabbit	Glass membrane feeder with pig intestine	28 °C	5–14	18–36	78–100%	6–64%	24–81%	Porcine stable-equine kidney cells	[93]
* Culex * * bitaeniorhynchus *	India	G-I (733913)	NA	ND	Viremic ducklings	NA	ND	9–12	1	9–100%	ND	100%	Transmission to ducklings	[102]
	India	G-I (733913)	NA	ND	Viremic chickens	NA	ND	10–12	24	47–62%	ND	64–89%	Transmission to chickens	[103]
	India	G-I (733913)	NA	ND	Viremic chickens	NA	ND	ND	ND	ND	ND	Successful	Transmission to chickens	[104]
* Culex * * fuscocephala *	Taiwan	ND (TaiAn 171)	NA	10^−0.89^–10^−1.91^ mouse LD50	NA	Viremic pigs	ND	12–21	ND	ND	ND	0–68%	Transmission to chickens	[105]
	Thailand, Chiengmai valley	ND (BKM-984-70)	NA	8 PFU per mosquito	NA	Viremic chicken	ND	10–27	ND	95–100%	ND	10–20%	Transmission to chickens	[106]
* Culex gelidus *	Australia, Cairns (wild)	G-II (TS3306)	C6/36 and porcine stable-equine kidney cells	10^4.5±0.1^ CCID_50_/mL	Heparinized rabbit	Glass membrane feeder with pig intestine membrane	28 °C	14–15	4	100%	ND	100%	Detection of virus in brain aspirates of recipient suckling mice	[93]
	US, Malayan strain	ND (FM380)	ND	ND	NA	Viraemic chicken	27 °C	6–21	4–43	ND	ND	8–63%	Development of encephalitis in laboratory-reared mice	[107]
* Culex pipiens *	China, Shangai	G-I (SH7), G-III (SH15)	C6/36 cells	4.9–8.3 log TCID_50_/mL	Defibrinated mice	Hemotek membrane feeding and cotton pledgets	ND	7–14	11–52	45%	30% ^◊^	23%	TCID_50_ assay on BHK-21 cells	[108]
	Pennsylvania, US	G-III (Nakayama)	C6/36 cells	8.1 log_10_ PFU/mL	Goose	Cotton pledgets	26 °C	14	5–50	10%	40%	0%	Vero cells	[84]
	UK, Liverpool	G-II (CNS138-11)	Vero cells	10^6^ PFU/mL	Heparinized human	Hemotek with collagen membrane	18 °C	21	18	100%	ND	72%	Semi-quantitative qPCR	[89]
*Culex pipiens molestus* (Forsskål, 1775)	Taiwan, Taipei	ND (SH)	C6/36 cells and suckling mice brains	5.54 log_10_ PFU/mL	Defibrinated rabbit	Hanging drop method	28–32 °C	14	3–5	ND	ND	91%	Inoculation of brain tissue aspirates from recipient mice on to C6/36 cells	[109]
	US, Oakland	G-III (Nakayama)	ND	ND	Defibrinated rabbit	Cotton pledgets	27 °C	7–20	1	ND	ND	22% ^#^	Development of encephalitis in laboratory-reared mice	[96]
	Tashkent, Uzbekistan	ND (ROK-2.0028)	Vero cells	10^4^ PFU/mL	NA	Viremic chicken	26 °C	16–27	13–53	47–56%	25–26%	8%	Vero cells	[110]
* Culex pipiens pallens *	Japan	G-III (JaGAr 01)	ND	ND	NA	Infected lizards	ND	ND	ND	ND	ND	Successful	Transmission from infected mosquitoes to uninfected lizards and from infected lizards to mice via mosquito	[111]
	Japan, Sapporo	G-III (JANAr-5681)	C6/36 cells	6.2 PFU/mL (blood) and 3.7 PFU/mL (chicken)	ND	Cotton pledgets or viremic chicken	20 or 28 °C	0–20	10	30%	ND	ND	BHK-21 cells and IFA	[98]
	Korea, Gyeonggi Province	ND (ROK-2.0028)	Vero cells	10^5.2^ PFU/mL	NA	Viremic chicken	26 °C	13–34	32	6%	0%	ND	Vero cells	[112]
* Culex pipiens pipiens *	France, Montpellier and Nice	G-III (RP-9) and G-V (XZ0934)	Chicken fibroblast-derived DF1 cells	8 × 10^6^ FFU/mL	Washed rabbit erythrocytes	Cotton pledgets	26 °C	7–13	5–20	70–92%	26–80% ^◊^	12–41%	BHK-21 cells	[87]
	UK, Caldbeck	G-III (SA14)	Vero cells	1.8 × 10^6^ PFU/mL	Defibrinated horse	Hemotek with parafilm membrane	20 and 25 °C	14	20–56	69–90%	12–70%	0–70%	RT-PCR and isolation in Vero cells	[15]
	US, Yakima	G-III (Nakayama)	ND	ND	Defibrinated rabbit	Cotton pledgets	27 °C	14–20	1–4	ND	ND	12% ^#^	Development of encephalitis in laboratory-reared mice	[96]
* Culex * * pseudovishnui *	India	G-III (P20778)	NA	ND	Viremic chicks	NA	ND	8	ND	ND	60%	75%	Transmission to chickens	[113]
	India	G-III (P20778)	NA	ND	ND	ND	ND	1–10	ND	ND	49%	51%	Antigen detection is mosquito heads resp. salivary glands	[114]
* Culex quinquefasciatus *	Australia, Mareeba (wild)	G-II (TS3306)	C6/36 and porcine stable-equine kidney cells	10^4.5±0.1^ CCID_50_/mL	Heparinized rabbit blood	Glass membrane feeder with pig intestine membrane	28 °C	14–15	27	56%	ND	0%	Detection of virus in brain aspirates of recipient suckling mice	[93]
	Australia, Gold coast	G-II (TS3306)	C6/36 and porcine stable-equine kidney cells	10^4.5±0.1^ CCID_50_/mL	Heparinized rabbit	Glass membrane feeder with pig intestine	28 °C	17/19	8–51	98%	28%	50%	Porcine stable-equine kidney cells	[93]
	New-Zealand, Wellington	G-III (Nakayama)	C6/36 cells	8.1 log_10_ PFU/mL	Goose	Cotton pledgets	24 °C	14	6–36	17%	0%	ND	Vero cells	[84]
	US, Rutgers	G-III (Nakayama)	C6/36 cells	8.1 log_10_ PFU/mL	Goose	Cotton pledgets	26 °C	14	43–50	86%	0%	0%	Vero cells	[84]
	Brazil	G-V (Muar)	Vero cells	4 log_10_ PFU/mL	Defibrinated horse	Hemotek with Parafilm membrane	23 and 28 °C	0–21	3–32	25–100%	21–70%	3–70%	Vero cells	[85]
	US	G-III (Nakayama)	ND	ND	Defibrinated rabbit	Cotton pledgets	27 °C	11–25	1–9	ND	ND	3% ^#^	Development of encephalitis in laboratory-reared mice	[96]
* Culex sitiens *	Australia, Coomera Islands	G-II (TS3306)	C6/36 and porcine stable-equine kidney cells	10^4.5±0.1^ CCID_50_/mL	Heparinized rabbit	Glass membrane feeder with pig intestine	28 °C	5–14	15–36	83–92%	6–33%	7–67%	Porcine stable-equine kidney cells	[93]
* Culex tarsalis *	US	G-II (Nakayama)	ND	ND	Defibrinated rabbit	Cotton pledgets	27 °C	6–10	1–12	ND	ND	1% ^#^	Development of encephalitis in laboratory-reared mice	[96]
* Culex * * tritaeniorhynchus *	Japan, Sapporo	G-III (JANAr-5681)	C6/36 cells	6.2 PFU/mL (blood) and 3.7 PFU/mL (chicken)	ND	Cotton pledgets or viremic chicken	20 or 28 °C	0–20	15	100%	ND	100%	BHK-21 cells and IFA	[98]
	Japan, Narita	G-I (17CxIT-I4-D31), 3 (JaGAr 01) and 5 (Muar)	C6/36 cells	8.9, 8.6, and 7.1 log_10_ FFU/mL	Defibrinated rabbit	Hemotek with pig intestine membrane	27° C	7–14	27–51	85–99%	81–96%	76–89%	RT-qPCR or Vero cells	[12]
	Korea, Gyeonggi Province	ND (ROK-2.0028)	Vero cells	10^4.3^ or 10^5.2^	NA	Viremic chicken	26 °C	13–34	10–18	100%	80–93%	50%	Vero cells	[112]
	Taiwan, Taipei	ND (SH)	C6/36 cells and suckling mice brains	5.48 log_10_ PFU/mL	Defibrinated rabbit	Hanging drop method	28–32 °C	14	6–8	ND	ND	100%	Inoculation of brain tissue aspirates from recipient mice on to C6/36 cells	[109]
* Culex vishnui *	India	G-III (P20778)	ND	ND	ND	Oral infection	ND	1–10	100	ND	34%	48%	Antigen detection in mosquito heads resp. salivary glands	[115]
*Culiseta annulata* (Schrank, 1776)	UK, Little Neston	G-II (CNS138-11)	Vero cells	10^6^ PFU/mL	Heparinized human	Hemotek with collagen membrane	21 and 24 °C	14–28	5–35	0–57%	ND	0–30%	Semi-quantitative qPCR	[89]
*Culiseta incidens* (Thomson, 1869)	US	G-III (Nakayama)	ND	ND	Defibrinated rabbit	Cotton pledgets	27 °C	8–14	1–22	ND	ND	5% ^#^	Development of encephalitis in laboratory-reared mice	[96]
*Culiseta inornata* (Williston, 1893)	US	G-III (Nakayama)	ND	ND	Defibrinated rabbit	Cotton pledgets	27 °C	10–20	2–12	ND	ND	4% ^#^	Development of encephalitis in laboratory-reared mice	[96]
*Opifex fuscus* (Hutton, 1902)	New-Zealand, Wellington	G-III (Nakayama)	C6/36 cells	10^8.1^	Goose	Cotton pledgets	24 °C	14	37–50	74%	70%	0%	Vero cells	[84]
* Verrallina funerea *	Australia, Cairns (wild s)	G-II (TS3306)	C6/36 and porcine stable-equine kidney cells	10^4.5±0.1^ CCID_50_/mL	Heparinized rabbit	Glass membrane feeder with pig intestine membrane	28 °C	14–15	36	11%	ND	7%	Detection of virus in brain aspirates of recipient suckling mice	[93]

* Infection rate = virus detected in mosquito body; ** Dissemination rate = virus detected in legs, wings, and/or mosquito heads, calculated on total number of mosquitoes, except when indicated with ◊ = dissemination rate calculated on total number of successfully infected mosquitoes; *** Transmission rates = virus detected in saliva and/or by letting infected mosquitoes feed on naïve animals; the Hemotek system is an artificial feeding system using an electric heating element to maintain the temperature of the blood meal at 37 °C; ND indicates lack of data in the given study; **^#^** = estimated percentages (minimum values) due to incomplete data in the given study; NA = not applicable; FFA = fluorescent foci assay; IFAT = indirect immunofluorescent antibody test; FFU = focus forming unit; PFU = plaque forming units; CCID_50_ = cell culture infectious dose 50% assay; TCID_50_ = tissue culture infective dose 50% assay; WMICLD50 = weanling mice intracranial lethal dose 50% assay.

**Table 4 pathogens-11-00317-t004:** Potential and confirmed vectors for JEV. Potential vectors are only proven competent in vector competence experiments while confirmed vectors are additionally found positive in the field. Most efficient confirmed vectors are based on the extent of their transmission rate (>70%) calculated in vector competence studies.

Mosquito Species	Potential Vectors	Confirmed Vectors	References
* Aedes albopictus *		X	[19,20,88,95,116]
* Aedes detritus *	X		[85]
* Aedes dorsalis *	X		[96]
* Aedes japonicus *	X		[12,97,98]
* Aedes kochi *	X		[93]
* Aedes nigromaculis *	X		[96]
* Aedes notoscriptus *	X		[93]
* Aedes vexans *		X	[19,20,99]
* Aedes vigilax *		X	[25,26,93]
* Anopheles tessellatus *		X	[19,100]
* Armigeres subalbatus *		X	[19,20,28,32,34,101]
* Culex annulirostris *		X	[39,93,99]
* Culex bitaeniorhynchus *		X	[21,28,43,44,45,102,103,104]
* Culex fuscocephala *		X	[19,22,27,31,46,48,49,50,105,106]
* Culex gelidus *		X	[22,23,27,31,46,50,51,52,53,54,55,56,57,58,59,60,61,93,107,115]
* Culex pipiens *		X	[15,43,44,62,63,64,84,87,89,96,108,109,110]
* Culex pipiens pallens *		X	[38,98,111,112]
* Culex pseudovishnui *		X	[37,65,66,67,68,113,114]
* Culex quinquefasciatus *		X	[19,20,22,28,31,69,70,84,85,93,96]
* Culex sitiens *		X	[20,21,24,26,51,70,71,72,73,93]
* Culex tarsalis *		X	[32,96]
* Culex tritaeniorhynchus *		X	[12,19,21,24,27,28,31,33,34,35,38,41,42,43,44,46,47,50,53,54,55,56,59,64,67,70,74,75,76,77,78,79,80,98,109,112]
* Culex vishnui *		X	[21,37,46,47,53,67,79,80,114]
* Culiseta annulata *	X		[89]
* Culiseta incidens *	X		[96]
* Culiseta inornata *	X		[96]
* Verrallina funerea *	X		[93]

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
