# Peer review of "Japanese Encephalitis Virus Interaction with Mosquitoes: A Review of Vector Competence, Vector Capacity and Mosquito Immunity"

_pathogens, 2022, doi:10.3390/pathogens11030317_

Round 1

Reviewer 1 Report

The manuscript entitled “Japanese encephalitis virus interaction with mosquitoes: a review of vector competence, vector capacity and mosquito immunity” summarizes information and data from current and past studies that focus on the potential for various species/populations as competent vectors of Japanese encephalitis virus throughout various parts of the world.

Overall, the manuscript is poorly written and very “wordy” and should be condensed. Some Tables are difficult to interpret. I have written some comments directly on the manuscript, but may be difficult to read since it is single spaced.

Specific comments:

  1. Introduction.
  2. Line 31. This sentence is awkward. Also, birds are considered to be the natural hosts, while swine are considered to be amplifying hosts.
  3. Line 48 and elsewhere. For numbers >1,000, use a comma. For decimals, use periods.
  4. Line 50. JEV is endemic in Southeast and East Asia, including the temperate zone of northeastern China, Japan and Korea.
  5. Line 56 and elsewhere. Delete “namely” and replace with “e.g.,”. Also, can replace “such as” with “e.g.,”.
  6. Lines 73-74. How does climate make mosquitoes more susceptible to JEV. Need to indicate what climatic factors/parameters may affect vector competency.
  7. Line 92-93 and elsewhere. Ensure that genus and species names are in italics.
  8. Table 1. This is very difficult to read. To shorten some parts, replace Genotype 1, etc. with “G-I”, “G-II”, etc. For Detection method, just use “isolation”, the assumption is that it is “virus isolation”. For Total # mosquitoes tested, just use “# tested”. Also, replace “not given” with “ND”. For the numbers, replace the period with a comma. As written for English, it looks like 3.95 Ae. butleri were tested, when it was actually 3,950 mosquitoes. For decimals, use periods.
  9. Lines 117-138. Ensure that genus and species are in italics as indicated above.
  10. Table 2. See comments above for Table 1.
  11. 2.1.1. Line 153-154. Move this sentence and other related information to before Table 4.
  12. Table 3. See comments above for Table 1. Shorten headings, e.g., “Country” rather than “Contry where study was conducted”. Detection method: shorten, e.g., replace “plaque assay in Vero cells” with “Vero cells”.
  13. Table 4. Aedes vexans is a complex of at least 3 species, with the type locality in Europe, and other species in East Asia and the US. Unfortunately, no one at this point has defined/described these species.
  14. Lines 178-182. Use italics for genus and species. This is very wordy and should be shortened. .
  15. Line 191 and elsewhere. Use “G-III” rather than “genotype 3”.
  16. Line 231. Transmission is not throughout the year in endemic temperate zones, e.g., Japan and Korea.
  17. Line 237. In Korea and Japan, JEV is associated with the migration of the primary vector (Cx. tritaeniorhynchus) and migratory reservoir bird populations, in addition to pigs that serve as amplification hosts.
  18. Line 248. A reason why Ae. Japonicas has not been found infected in the wild, is that it is primarily a forest mosquito where large water birds don’t frequent. So vector habitat is very important.
  19. Lines 257-261. It is crucial that there are vector-host interactions. If these do not overlap, then they don’t serve as very good vectors even if they are competent.
  20. Line 271 and elsewhere. Rather than “spread”, use dissemination.
  21. 2.2.1. This section is very long and perhaps not the rational and explanation can be greatly shortened. Hypothetical reasons for the various barriers are described in excess. Some parts appear to be repeated.
  22. Lines 333-344. Parts of this is related to vector-host contact. This should be addressed. For example, what is the potential for Ae. aegypti and Ae. japonicas to interact with reservoir bird populations.
  23. 2.2.2. This is likely too much information for the paper and parts could be deleted.
  24. 2.2.4. This could be shortened to one paragraph.
  25. 3. Conclusion. Too long and should be shortened to highlight the important aspects of the review.
  26. References. The references need to be carefully checked, e.g., scientific names are not in italics, titles are capitalized, rather than in lower case, journals are not listed, reference 132 is Kim HC, not ChulKim H, when specific districts are used, then they should be in caps (e.g., 141, “Cuddalores Disrict).

Reviewer 2 Report

Please find in attachment my remarks and suggestion

Reviewer 3 Report

1.    Dot should be added before “Not….”on line 170.
2.    Little open circles under number (57-100ï¼…, 30ï¼… and 26-80ï¼…) should be deleted on ï¼… disseminated infection of Table 3.
3.  Spacer between character and dot should be revised on line 114, 139 and 166.
4.  Italic of species name should be make sure on line 178-180 and 333-334. 

Round 2

Reviewer 1 Report

The manuscript is relatively well organized and changes have been made appropriately.

However, there are minor grammatical issues, e.g., using a comma before, e.g., some spacing issues, and reference citations that need to be modified, e.g., scientific names in italics and lower case for reference titles.

In addition, in the Table1, under MIR, there is a "/" with no designation. It might be assumed that this means that there is "no data". This should be explained in a footnote, or just put "ND". Similarly for Table 3, under "% disseminated" and "% transmission".

For Table 4, in most cases the species doesn't need to be hyphenated as the species name fits under the Genus.

Reference 98. Check the reference. This should probably be "Japanese encephalitis", not "Japan encephalitis".
